# Addressing disparity in attitudes and utilization of family planning among married couples in the pastoralist community of Fentale District, Eastern Ethiopia

**Sena Adugna Beyene** [1,2] *, **Sileshi Garoma**[3], **Tefera Belachew**[4]

1 Institute of Health, Department of Population and Family Health, Jimma University, Jimma, Ethiopia, 2 Department of Statistics, College of Natural Sciences, Jimma University, Jimma, Ethiopia, 3 Departments of Public Health, Adama Hospital Medical College, Adama, Ethiopia, 4 Department of Nutrition & Dietetics, Institute of Health, Jimma University, Jimma, Ethiopia

* senaada491@gmail.com

## Abstract

### Background

Despite progress in national reproductive health programs, pastoralist regions like Fentale District in Eastern Ethiopia face challenges with low contraceptive coverage, attributed to insufficient positive attitudes and uptake among couples.

### Methods

This cross-sectional study was conducted from October 1 to December 25, 2021, in Fentale District, Eastern Ethiopia. It involved 1,496 couples selected using multistage sampling. Data were entered into EPI Data and analyzed with SPSS (v23.0) and STATA (v14.0), employing descriptive statistics, bivariate analysis, and binary logistic regression to identify predictors of contraceptive attitudes and use.

### Results

The study's response rate was 93.8%, with 1,404 pastoralist couples participating, equally split between women and men. A nomadic-pastoralist lifestyle was common (64.6%), and family planning discussions were rare (93.2%). Gender disparities in contraceptive attitudes and use were evident. Contraception use was reported by 27.4%, with women (41.2%) out-numbering men (13.5%). Women showed more positive attitudes towards contraception (87.9% vs. 31.9% for men). Only 33% had favorable attitudes towards different contraceptive methods, with women more likely to be positive. Modern contraception use was low (18.2%), with women (34.8%) predominating over men (1.7%). Among users, women had a more favorable attitude (78.5% vs. 6.6% for men). Health extension workers were key information providers. Predictors of contraceptive attitudes and use included sex, education, occupation, electronic device ownership, migration frequency, treatment preferences, and family planning discussions.

**Data Availability Statement:** The primary results and analyses are presented directly within the main text of the manuscript. This includes all key data points, statistical analyses, and relevant figures or

tables that are crucial to the understanding and validation of our findings. For any further information or specific inquiries about the data, readers are encouraged to contact the corresponding author, whose details are provided in the manuscript.

**Funding:** The author(s) received no specific funding for this work.

**Competing interests:** The authors declare that they have no conflicts interest.

**Abbreviations:** AOR, Adjusted Odds Ratio; COR, Crude Odds Ratio; CHW(s), Community Health Worker(s); CI, Confidence Interval; FP, Family Planning; GEE, Generalized Estimating Equations; HCP(s), Health Care Provider(s); STATA, Stata is statistical software for data science; SPSS, Statistical Package for Social Sciences; SDGs, Sustainable Development Goals; SSA, Sub-Saharan Africa; IPTW, Inverse Probability of Treatment-Weighted; HSTP, Health Sector Transformation Plan of Ethiopia; MC, Modern Contraceptive.

## Conclusion

The limited positive attitude towards and use of family planning in Fentale District may stem from unfavorable attitudes, low adoption, and couple disparities. Key factors include gender, education, occupation, electronic device ownership, migration, treatment preferences, and family planning discussions. Targeted educational campaigns for men are needed to address these issues and reduce the gender gap in contraceptive attitudes and use.

## 1. Introduction

Family planning is a foundational element of global health strategies, closely linked with the Sustainable Development Goals (SDGs) that aim to improve the health and well-being of women, children, and adolescents globally [1, 2]. It is a critical tool in tackling complex issues such as poverty reduction, socioeconomic development, gender equality, and the lowering of maternal and child mortality rates, ultimately driving socio-economic progress and enhancing health outcomes [3, 4].

Despite the clear advantages of family planning, there are significant disparities in its availability and use across the world [5]. In many developing nations, including Ethiopia, certain populations face substantial hurdles in accessing and utilizing family planning services. Pastoralist communities, with their nomadic lifestyle and remote locations, often find themselves beyond the reach of standard health services [3].

The pastoralist community in Fentale District, Eastern Ethiopia, is a case in point. These communities, whose livelihoods depend on livestock, face unique obstacles in accessing family planning information and resources [6]. Their mobile lifestyle, along with cultural practices and limited education, results in significant gaps in the use of family planning methods [7].

In Ethiopia, the ongoing health challenges underscore the need for effective family planning interventions to meet international health commitments [8]. Sub-Saharan Africa, including Ethiopia, continues to battle high rates of maternal and child mortality, calling for targeted efforts to improve health outcomes [9]. Despite some progress, many Ethiopian women still have unmet family planning needs, with disparities evident across economic, cultural, educational, and geographic lines [10–12]. Pastoralist populations, among other marginalized groups, face specific challenges that require tailored interventions [13–15].

While Ethiopia's overall family planning indicators show progress, pastoralist communities lag behind national averages [16]. Addressing these disparities involves understanding and addressing gender norms, traditional practices, and access barriers limitations [17]. Education that targets both husbands and wives in pastoralist communities is essential, highlighting the importance of involving both partners in decision-making. Strengthening male participation is in line with broader public health goals and supports collaborative efforts for better health outcomes [18].

The study is in harmony with global health initiatives and Ethiopia's Health Sector Transformation Plan II (HSTP II), with a focus on modern contraceptive use and attitudes attitudes [17]. We investigate the various aspects of male involvement in family planning, acknowledging its key role in shaping reproductive health behaviors outcomes [19]. The recognition of male involvement in reducing maternal and infant health issues has grown since the ICPD in Cairo [20]. Yet, past studies often focused on women's disadvantages without considering men's roles, leading to incomplete understandings and neglect of gender inequities and violence in reproductive health [21].

In Fentale District, male dominance is evident, particularly in decisions affecting their wives [22, 23]. Women are generally limited in their participation in social activities and travel, except for specific local tasks [24]. However, men can act as conduits for information when they travel; potentially influencing family planning decisions [25]. Pastoral women, contrary to common perceptions, have significant informal power within their households and communities [26]. The extent of this power is influenced by various factors, including a woman's age, her husband's social standing, the number and age of her sons, and, in some societies, her eloquence and wisdom (Watson and C, 2010). The relationship with other family members also plays a role, making women-headed households potentially less socially powerful [24]. Reproductive health is considered a private matter, usually discussed only between the husband and wife [27]. In the pastoralist community, cultural influences place decision-making power in the hands of husbands, making it hierarchical [28].

The study aims to explore attitudes toward contraception and the factors influencing its use, providing essential insights for effective interventions [29]. We stress the importance of considering sociocultural factors and developing strategies tailored to pastoralist communities like Fentale District [30]. Through our analysis, we identify key factors affecting family planning outcomes, highlighting the need for adaptive social services and mobile clinics to reach nomadic populations [31]. Couple-based health education, with a focus on male involvement, appears as a promising strategy to enhance family planning practices in pastoralist settings [3].

The study seeks to address the disparities in family planning attitudes and use among married couples in Fentale District [32]. It is unique in its focus on the pastoralist population and its inclusion of male partners in family planning education [33].

By engaging both husbands and wives, we aim to fill knowledge gaps and promote a more inclusive approach to family planning [34]. This research is vital in extending the benefits of family planning to all, including marginalized groups, and in contributing to improved reproductive health and sustainable development in Ethiopia [8].

## 2. Methods and materials

### 2.1. Study area

This research is situated in Fentale Woreda, located within the East Showa zone of the Oromia regional state, situated in the southern part of the northern rift valley of Ethiopia. The geographical landscape is characterized by a nomadic lifestyle, Agro-Pastoralism, and distinctive seasonal migration patterns typical of pastoralist populations. The local economy revolves around livestock production, serving as a central pillar. Fentale district, situated in the East Showa Zone in Fantalle Woreda, comprises 20 kebeles, including 18 rural and 2 urban administration kebeles., with 15 specifically designated as pastoralist kebeles. The pastoralist villages selected for this study include Kobo, Benti, Gola, Dhaga Edu, Tututi, Ilalla, and Gelcha. Healthcare infrastructure in the district consists of four health centers, each village hosting a health post, and an additional health center located in Metehara City. Additionally, a hospital, primarily serving non-pastoralists and the staff of the Metehara sugar factory, is present. Importantly, Family Planning services are exclusively provided at the Metehara Hospital and Health Center, posing a significant challenge for pastoralists due to the absence of direct transportation options. Accessing healthcare services involves an arduous journey of hours or even a full day of walking, leading to substantial delays for pastoralists waiting for transportation, which could extend up to a week. Staffing at the health centers is predominantly composed of nurses, with Health Extension Workers (HEWs) managing health posts. Traditional birth attendants, locally known as "Deesisttu Aadaa", play a pivotal role in assisting women during childbirth and hold significant respect within the community. The social, economic, and

cultural dimensions of the study area intricately contribute to its unique context, shaping the backdrop against which this research unfolds. For a more detailed exploration of these aspects, refer to the comprehensive description provided in the referenced source, offering a holistic perspective on the study setting [35, 36].

## 2.2. Study design, time frame, and sampling approach

Fig 1 illustrates the multi-stage sampling strategy employed in this study. Districts (woredas) were used as primary sampling units (PSUs) and sub-districts (kebeles) as secondary sampling units (SSUs). Fentale district, located in the East Showa Zone, comprises 20 kebeles, including 18 rural and 2 urban administration kebeles. Initially, 15 pastoralist kebeles were purposively selected based on criteria such as accessibility, social structure, economic strength, and pastoralist characteristics. From these, seven kebeles were randomly chosen.

The source population consisted of married couples, with women of reproductive age being systematically sampled from each kebele. This systematic sampling approach ensured representative selection, and villages were selected based on their proximity to health facilities to reduce variability. In the seven selected pastoralist sub-districts, there were 1,045 women of reproductive age and their husbands, totaling 2,090 married couples across 5,223 households.

The sample size allocation was proportional to the number of households in each kebele. Within each sub-district, married couples were selected for interviews at equal intervals using a systematic sampling technique, as previously detailed in [3]. Fieldwork for this couple-based cross-sectional study was conducted from October 1 to December 25, 2021 (Fig 1).

## 2.3. Study population

This study concentrated on the pastoralist community living in Fentale Woreda, situated within the East Showa zone of the Oromia regional state in Ethiopia. The target population was meticulously defined with specific inclusion and exclusion criteria to highlight the distinctive attributes of the participants. The source population for this research comprised the pastoralist community in Fentale Woreda, known for their nomadic or semi-nomadic lifestyle, which revolves around livestock herding and migrating in search of pasture and water for their animals.

From this source population, the study population was selected based on the following criteria: Inclusion criteria included married women aged 15 to 49 and their husbands, non-pregnant women and their husbands, legally wedded couples who had lived in the village or mobile areas for at least a year due to livestock needs, couples cohabiting in the study area or mobile regions, couples committed to staying in the district or mobile areas for at least a year and a half from the data collection period, husbands consenting to their wives' participation, mentally competent couples, and husbands in monogamous marriages. For wives under 18, written informed consent was obtained from their husbands, adhering to cultural norms.

Exclusion criteria were applied to married women and their husbands outside the 15 to 49 age range, couples not legally married or where the husband refused to include his wife, mentally incapacitated couples, husbands in polygamous marriages, pregnant women and their husbands, couples not residing in the village or mobile areas for the past year, couples not cohabiting in the same household in the study area or mobile regions, and couples not intending to remain in the district or mobile areas for at least a year and a half from the time of data collection. These exclusions were aimed at maintaining the study's focus and reducing unnecessary data.

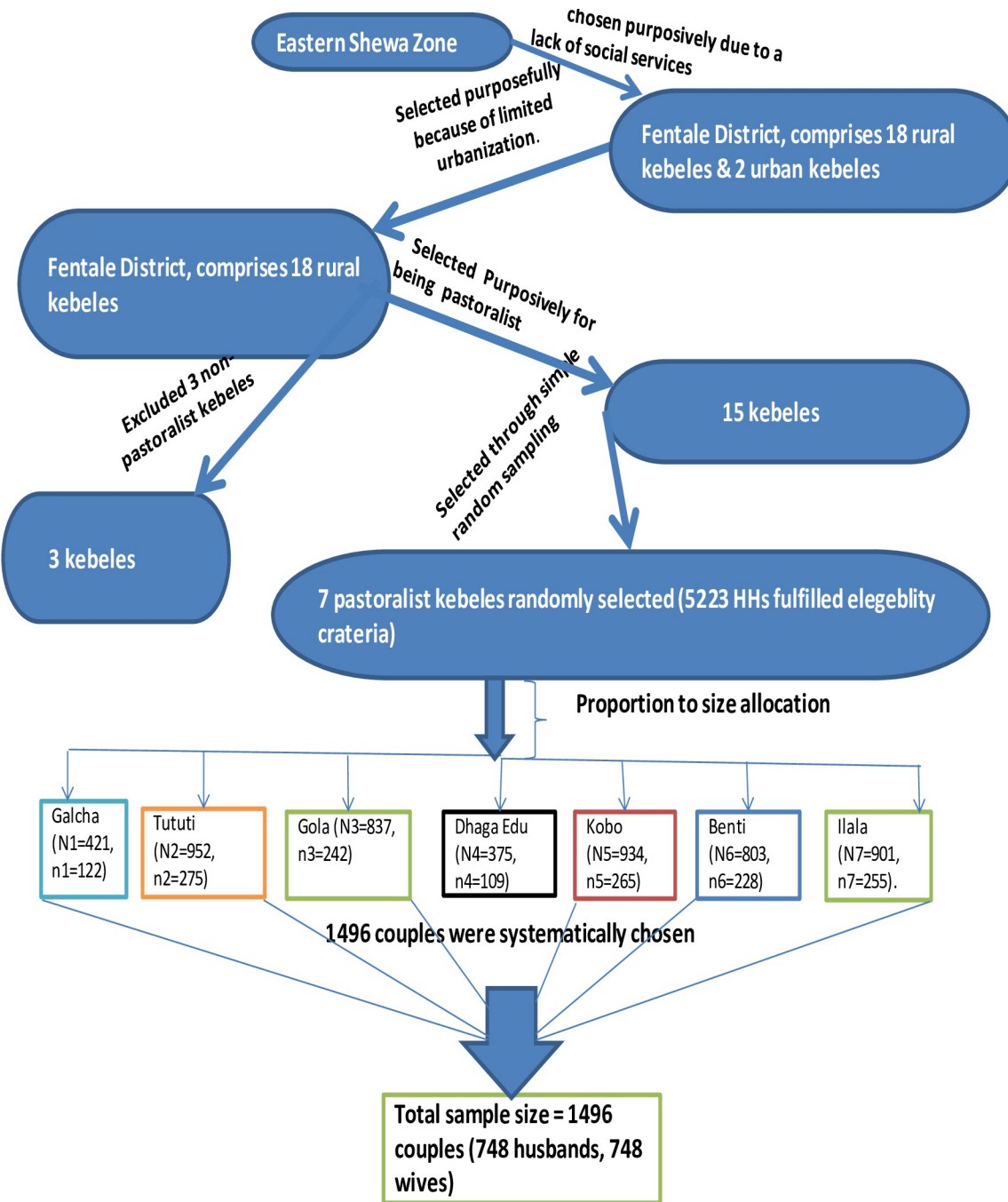

**Fig 1. Illustrates the schematic presentation of the sampling procedure conducted among married couples in the pastoralist community of Fentale District, Eastern Ethiopia, from October 1 to December 25, 2021.**

## 2.4. Sample size determination

The sample size determination was guided by specific parameters outlined in [3]. This involved meticulous calculation incorporating a standard normal deviation (Z) set at Zα/2 = 1.96 for a 95% confidence interval (CI) and assumed 90% power. The estimated key proportion of current family planning utilization in the Oromia regional state, based on the

EDHS (2016) report [37], served as a basis for the calculation (28.1% or 0.281). Accounting for a desired precision of 0.05, a finite population correction, and an Intra-Cluster Correlation Coefficient (ICC) variation of 0.05 for clustering effects. Initially, the sample size was determined to be 310 couples, with adjustments introducing a 20% increment to compensate for expected non-responses. Additionally, a design effect of 2.2 was applied. Consequently, the final sample size was determined to be 374 couples, resulting in a total of 748 couples per group selected using systematic sampling. The cumulative sample size for the study thus comprised 1496 couples. This systematic approach ensures the robustness and statistical significance of the study findings.

## 2.5. Data collection

The structured and pre-tested questionnaire, initially prepared in English, underwent translation into the local language (Oromiffa) and back into English to ensure question consistency. Separate questionnaires were administered for male and female respondents, covering socio-demographic and economic variables, reproductive history, and knowledge and practices related to family planning (FP). The questionnaire included information on ever use of contraceptives, current use, and reasons for not using contraceptives, as well as details on types of contraceptives, knowledge about usage, sources of family planning services, and potential side effects. Survey instruments were developed from a validated questionnaire, recognized as valid and reliable by experts [3, 25, 38–43]. Prior to main data collection, the questionnaire underwent pre-testing among 5% of couples in a different district to assess validity and clarity. The study involved 15 male and 15 female data collectors supervised by six field coordinators recruited from the local community. Male data collectors were matched with husband respondents, while female data collectors were paired with spousal respondents, a measure taken due to the sensitivity of the topic. Interviews were conducted individually with each couple in private settings to uphold confidentiality, with locations chosen based on participant comfort levels. Both data collectors and supervisors underwent a thorough six-day training session covering study objectives, procedures, data collection techniques, interviewing skills, and addressing any concerns. Practical exercises, such as role-playing, were included in the training curriculum. To ensure consistency and comparability, equivalent questionnaires were administered to both men and women, facilitating direct and accurate comparisons. Participants were provided with comprehensive information about the study, including its purpose, procedures, potential risks, and benefits, ensuring informed consent. Consistency in questionnaire administration across genders and the chosen analytical approach ensured a robust analysis, taking into account within-couple dynamics and utilizing culturally tailored data collection tools.

## 2.6. Data management and analysis approach

In the Data Management and Analysis Approach, rigorous procedures were followed to ensure accuracy and reliability of the collected data. Upon completion of questionnaire responses, each was meticulously checked for completeness and assigned unique codes. Data entry was performed using a validated template, which underwent thorough validation with 30 questionnaires, overseen by the principal investigator. Subsequently, the data were transferred to statistical software packages, including SPSS version 23.0 and STATA version 14.0, for comprehensive analysis. Frequency analyses were conducted to examine event occurrences, and specialized tests such as Pearson's Chi-square were utilized to determine significance. Associations between family planning attitude, contraceptive use, and various factors were explored using Pearson's Chi-square and logistic regression. Multicollinearity was assessed

using the Variance Inflation Factor (VIF), confirming the integrity of the data structure with no significant multicollinearity present. Only predictors showing a significant difference between women and men, with a probability value less than 5% were included in the multivariate logistic regression. This allowed for assessing their relationship with the outcome while controlling for other variables in the model. Multivariate binary logistic models were employed to predict factors influencing knowledge reporting odds ratios with confidence intervals for nuanced interpretation. Univariate analysis provided insights into the distribution of individual variables, while descriptive statistics summarized variable characteristics. Bivariate analysis examined relationships between variables through cross-tabulations and correlation analyses, identifying associations and dependencies. Crude odds ratios were calculated to gauge the association between individual independent variables and dependent variables, with adjusted odds ratios derived from multivariate binary logistic models. The study also considered survey design effects to ensure analytical validity. Results were meticulously reviewed to confirm the effectiveness of the chosen methods in addressing research questions and objectives while adapting to changing data characteristics.

## 2.7. Ethical consideration

All experimental protocols conducted in this study received approval from the Institutional Review Board of Jimma University and the Oromia Regional Health Bureau before the research commenced. Ethical clearance was obtained from the Institute of Health Institutional Review Board at Jimma University (IHRPG-927/2020) and the Health Ethical Review Committee of the Oromia Regional Health Bureau (BEFO/HBTFH/999/2020). Formal permission was also secured from the Health Department of the Eastern Shewa Zone in the Oromia regional state and Fentale District within Eastern Shewa Zone. Prior to their participation, all respondents provided informed, voluntary, written, and signed consent after receiving detailed explanations about the study's objectives and procedures. The research adhered strictly to fundamental principles of human research ethics, including respect for individuals, beneficence, voluntary participation, confidentiality, and justice. Additionally, written informed consent was obtained from spouses acting on behalf of wives under 18, in accordance with cultural norms and ethical guidelines. No direct compensation was offered, and all procedures followed regulations stipulated by the ethics committee.

## 2.8. Economic status measurement

While direct measures of economic status, such as household income or wealth index, were not included in the current study, we used several proxy indicators to capture the economic conditions of the participants. These proxies include possession of a radio, possession of a mobile phone, possession of a bank account, and access to the internet. These variables are often used as indicators of economic status in resource-limited settings, where direct income data might be challenging to collect accurately. Collecting direct income data in pastoralist communities can be particularly difficult due to factors such as nomadic lifestyle and lack of formal financial records [44]. Given these challenges, researchers often rely on more easily observable and less sensitive proxy indicators, such as possession of certain assets or access to services, to infer economic status in pastoralist communities [44].

## 2.9. Measurement of variables

**Dependent variable.** *Attitude towards family planning.* Twenty questions were asked towards modern contraceptive methods to assess participants' attitude. All the questions were in the form of a Likert scale. Thus, in binary form were recoded all the questions too, and

"agree" response was coded as "1" and "disagree" response was coded as "0"for the answers to each question. Then, the score was calculated for each participant between 0 and 20. Finally, the score was checked for normality a composite attitude variable was then created from the score using the mean as the cutoff score. Participants with mean scores or above were classified as "favorable attitude,", and those with below mean scores were classified as an "unfavorable attitude." Also, Internal consistency was properly measured to assess reliability, with Cronbach's Alpha (α) values indicating excellent internal consistency reliability for the attitude scale used in the study. The alpha (α) values for the 20 attitude-related items were very strong, measuring at α = 0.953.

*Family planning utilization*. Utilization was measured by the use of any family planning methods, categorized into users and non-users. Participants were queried about their contraceptive usage during the data collection period, and information was collected on current use of family planning methods. The assessment focused on the actual practice or use of family planning methods, determined by the question: "Are you or your partner currently using any method to delay or prevent pregnancy?" Details regarding the specific type of family planning method utilized were gathered, including female sterilization, male sterilization, pill, IUD, implant, injectable, male condom, female condom, periodic abstinence, lactational amenorrhea (LAM), withdrawal, other methods, and no method (considered as non-use of family planning methods). Data were collected from both the woman and her husband.

**Independent variables..** *Socio-demographic characteristics*.

○ **Age Distribution:** Measured as the age of the participants at the time of the survey.

○ **Gender Distribution:** Recorded as male or female.

○ **Age at First Marriage:** Measured as the age at which participants first got married.

○ **Religion:** Categorized as Muslim or Christian.

○ **Ethnicity:** Predominantly categorized as Oromo.

○ **Educational Status:** Categorized as no formal education, primary education, and secondary education or above.

○ **Occupational Status:** Categorized into groups such as nomadic pastoralists, business owners, students, daily laborers, employed individuals, and agro-pastoralists.

○ **Family Size:** Measured as the total number of people in the household.

○ **Desired Number of Children:** Measured through direct questioning about the ideal number of children desired.

○ **Need for a Future Child:** Assessed by asking participants about their desire for more children in the future.
   **Economic proxies**.

○ **Possession of a Radio:** Measured as a binary variable (yes/no).

○ **Possession of a Mobile Phone:** Measured as a binary variable (yes/no).

○ **Possession of a Bank Account:** Measured as a binary variable (yes/no).

○ **Internet Access:** Measured as a binary variable (yes/no).
   **Social variables**.

○ **Media Exposure:** Frequency of exposure to various media sources (e.g., radio, TV, internet) was recorded.

○ **Frequency of Migration:** Measured as the number of times participants migrated in a given period.

○ **Migration Destination:** Categorized based on whether migration occurred within the Fentale District or outside.

○ **Family Structure Who Migrates Mostly:** Identified the primary individual(s) who migrate within the family.

○ **Where Couples Seek Treatment:** Categorized based on the type of health facility (e.g., health centers, traditional healers, Religious places).
  **Access to health services**.

○ **Distance from Health Center:** Measured as the time taken to reach the nearest health center, categorized into less than one hour or more than one hour.

○ **Discussion of Family Planning:** Measured by asking participants if they have ever discussed family planning with their spouse (yes/no).

## 3. Results

### 3.1. Investigating the relationship between socio-demographic characteristics, economic factors, social elements, health service accessibility, and family planning attitudes

This study, conducted in the Fentale District of Eastern Ethiopia, examined family planning attitudes among pastoralist communities. Out of 1496 eligible married couples, 93.8% (1404) participated in the survey. The nomadic lifestyle of these pastoralists, involving regular movement for livestock, presented challenges in engaging participants. The sample was evenly split between men and women (50.0% each), with most couples marrying at 18 or older (40.0%). The most represented age group was 25–29 years (26.5%). Most couples lacked formal education (53.8%) and were Muslim (97.9%) of Oromo ethnicity (99.6%). Nomadic pastoralism was the predominant occupation (64.6%). Few owned a radio (5.7%) or a mobile phone (18.2%) and many lacked a bank account (58.2%) or internet access (94.2%). Media exposure was low (12.9%). Most migrated four times or more (34.4%) within the Fentale District (85.9%), with the household head usually migrating (45.7%). Despite living over an hour from a health center (57.5%), most sought treatment there (43.3%). Few had discussed family planning (6.8%), and most had large families (51.4%) and desired more children (83.3%).

The study found significant differences in family planning attitudes between men and women (p < 0.05), with women more favorably inclined (64.1% vs. 35.9% for men). Age at first marriage and different age groups did not significantly affect attitudes (p = 0.841 and p = 0.095, respectively). However, education level did (p < 0.05), with more educated couples showing more favorable attitudes. Muslims had slightly less favorable attitudes than Christians (p = 0.045), but ethnicity did not significantly impact attitudes (p = 0.394). Occupation was significant (p < 0.05), with nomadic pastoralists more favorably disposed than others, especially business owners. Ownership of a radio, mobile phone, bank account, internet access, and media exposure were all associated with more favorable attitudes (p < 0.05). Migration frequency and destination, as well as who in the family migrates, were significant (p < 0.05, p = 0.002). Couples migrating once or with all family members migrating had the most favorable attitudes. Treatment-seeking location was significant (p < 0.05), with those closer to health centers and seeking treatment there having more favorable attitudes. Discussing family

planning correlated with more favorable attitudes (p < 0.05). Desired number of children and the need for more children were significant (p < 0.05), but family size was not (p > 0.05).

These findings reveal the intricate role of socio-demographic factors in family planning attitudes among married couples in Fentale, guiding targeted interventions to improve attitudes and family planning service utilization (See Table 1 for detailed results.)

## 3.2. Variations in attitudes and perceptions regarding family planning among married couples

Table 2 Attitudes and perceptions of married couples toward Family Planning (FP) services were investigated, focusing on susceptibility, severity, benefits, barriers, beliefs, and self-efficacy, with a gender breakdown. Key findings include:

Perceived Susceptibility of Family Planning (FP) Services:

- Approximately half of the respondents felt vulnerable to unwanted pregnancy without contraception, with a higher perception among women (42.9%) than men (25.6%).

- Moreover, a significant portion perceived a risk of maternal mortality associated with unwanted pregnancy, particularly pronounced in women (30.1%) compared to men (20.9%).

Perceived Severity of Family Planning (FP) Services:

- Among 1404 married couples, a notable percentage believed that an unwanted pregnancy could lead to unsafe abortion, with higher rates among women (28.1%) than men (8.3%).

- The perception that unsafe abortion could result in maternal death was significantly higher among women (36.9%) than men (11.3%).

Perceived Benefit of Family Planning (FP) Services:

- Only a fraction of respondents perceived the effectiveness of modern contraceptives in preventing pregnancy, notably higher for women (35.5%) than men (23.2%).

- Overall, a considerable proportion of married couples perceived that modern contraceptives prevent the consequences of abortion, with no significant difference between men (36.5%) and women (36.6%).

- A minority perceived the benefits of certain modern contraceptives in preventing HIV/AIDS transmission, with more positive attitudes among men (17.7%) than women (12.5%).

Perceived Barrier of Family Planning (FP) Services:

- Less than one-third of couples perceive access to modern contraceptives as a significant barrier, more prevalent among women (39.7%) than men (19.1%).

- Additionally, a significant portion of married couples find the cost of using modern contraception affordable, with a notable difference between men (46.6%) and women (34.6%).

Beliefs about Family Planning (FP) Services:

- Results indicate that a substantial portion of married couples believe that using modern contraceptives does not contradict their religious principles, with a higher proportion among women (25.4%) compared to men (16.0%).

- Similarly, a significant percentage of married couples believe that using modern contraceptives aligns with their cultural principles, with women (34.5%) outnumbering men (28.5%).

**Table 1. Sample characteristics of married couples' attitudes towards family planning in the Fentale District, Eastern Ethiopia, covering the period from October 1 to December 25, 2021.**

| Characteristics | Total (n = 1404) | Attitude towards Family Planning | | P-value |
|---|---|---|---|---|
| | | *Favorable attitude N = 463* | *Unfavorable attitude N = 941* | |
| | **Frequency (%)** | *Frequency (%)* | *Frequency (%)* | |
| *Attitude towards Family Planning* | | | | *0.000** |
| *Unfavorable attitude* | **941 (67.0)** | | | |
| *Favorable attitude* | 463 (33.0%) | | | |
| *Sex* | | | | *0.000** |
| *Women* | *702(50.0)* | *297(64.1)* | *405(43.0)* | |
| *Men* | *702(50.0)* | *166(35.9)* | *536(57.0)* | |
| *Age at first Marriage* | | | | *0.841* |
| *< = 15 years* | *650(46.3)* | *219(47.3)* | *431(45.8)* | |
| *16–17 years* | *193(13.7)* | *61(13.2)* | *132(14.0)* | |
| *> = 18years* | *561(40.0)* | *183(39.5)* | *378(40.2) =* | |
| *Age* | | | | *0.095* |
| *15–19 years* | *109(7.8)* | *48(10.4)* | *61(6.5)* | |
| *20–24 years* | *308(21.9)* | *109(23.5)* | *199(21.1)* | |
| *25–29 years* | *372(26.5)* | *120(25.9)* | *252(26.8)* | |
| *30–34 years* | *280(19.9)* | *89(19.2)* | *191(20.3)* | |
| *35–39 years* | *119(8.5)* | *39(8.4)* | *80(8.5)* | |
| *40–44 years* | *117(8.3)* | *31(6.7)* | *86(9.1)* | |
| *> = 45 years* | *99(7.1)* | *27(5.8)* | *72(7.7)* | |
| *Educational status* | | | | *0.000** |
| *No formal education* | *756(53.8)* | *549(58.3)* | *207(44.7)* | |
| *Primary* | *458(32.6)* | *284(30.2)* | *174(37.6)* | |
| *Secondary & above* | *190(13.5)* | *108(11.5)* | *82(17.7)* | |
| *Religion* | | | | *0.045** |
| *Muslim* | *1374(97.9)* | *448(96.8)* | *926(98.4)* | |
| *Christian* | *30(2.1)* | *15(3.2)* | *15(1.6)* | |
| *Ethnicity* | | | | *0.394* |
| *Oromo* | *1398(99.6)* | *462(99.8)* | *936(99.5)* | |
| *Others* | *6(0.4)* | *1(0.2)* | *5(0.5)* | |
| *Occupational status* | | | | *0.000** |
| *Nomadic-pastoralist* | *907(64.6)* | *249(53.8)* | *658(69.9)* | |
| *Business* | *100(7.1)* | *40(8.6)* | *60(6.4)* | |
| *Others* | *25(1.8)* | *16(3.5)* | *9(1.0)* | |
| *Student* | *31(2.2)* | *16(3.5)* | *15(1.6)* | |
| *Agro-Pastoralist* | *341(24.3)* | *142(30.7)* | *199(21.1)* | |
| *Possession of radio* | | | | *0.000** |
| *No* | *1324(94.3)* | *420(90.7)* | *904(96.1)* | |
| *Yes* | *80(5.7)* | *43(9.3)* | *37(3,9)* | |
| *Possession of mobile phone* | | | | *0.000** |
| *No* | *1149(81.8)* | *350(75.6)* | *799(84.9)* | |
| *Yes* | *255(18.2)* | *113(24.4)* | *142(15.1)* | |
| *Possession of Bank account* | | | | *0.000** |
| *No* | *817(58.2)* | *319(68.9)* | *498(52.9)* | |
| *Yes* | *587(41.8)* | *144(31.1)* | *443(47.1)* | |
| *Use the internet* | | | | *0.000** |

*(Continued)*

**Table 1.** (Continued)

| Characteristics | Total (n = 1404) | Attitude towards Family Planning | | P-value |
|---|---|---|---|---|
| | | Favorable attitude N = 463 | Unfavorable attitude N = 941 | |
| | Frequency (%) | Frequency (%) | Frequency (%) | |
| No | 1322(94.2) | 424(91.6) | 898(95.4) | |
| Yes | 82(5.8) | 39(8.4) | 43(4.6) | |
| Couple's Exposure to media | | | | 0.000* |
| Less Frequent | 1223(87.1) | 380(82.1) | 843(89.6) | |
| More Frequent | 181(12.9) | 83(17.9) | 98(10.4) | |
| Frequency of Migration | | | | 0.000* |
| Once | 21(1.5) | 15(3.2) | 6(0.6) | |
| Twice | 247(17.6) | 128(27.6) | 119(12.6) | |
| Three | 388(27.6) | 135(29.2) | 253(26.9) | |
| Four | 483(34.4) | 133(28.7) | 350(37.2) | |
| Five& more | 265(18.9) | 52(11.2) | 213(22.6) | |
| Migration destination | | | | 0.002* |
| Within Fentale District | 1206(85.9) | 400(86.4) | 806(85.7) | |
| Outside Fentale District | 198(14.1) | 63(13.6) | 135(14.3) | |
| Family structure who migrate mostly | | | | 0.002* |
| Head of the household | 642(45.7) | 202(43.6) | 437(46.4) | |
| All family members | 389(27.7) | 46(9.9) | 145(15.4) | |
| Young Men | 373(26.6) | 215(46.4) | 359(38.2) | |
| Treatment Seeking | | | | 0.000* |
| At Health Sectors | 331(23.6) | 235(50.8) | 96(10.2) | |
| Traditional healers | 608(43.3) | 143(30.9) | 466(49.5) | |
| Religious places | 445(31.7) | 83(17.9) | 362(38.5) | |
| Others | 20(1.4) | 2(0.4) | 17(1.8) | |
| Distance from health center | | | | 0.000* |
| < 1 hour | 597(42.5) | 240(51.8) | 357(37.9) | |
| ≥ 1 hour | 807(57.5) | 223(48.2) | 584(62.1) | |
| Couple discussion of FP | | | | 0.000* |
| Never discussed | 1308(93.2) | 407(87.9) | 901(95.7) | |
| Discussed | 96(6.8) | 56(12.1) | 40(4.3) | |
| Family size | | | | 0.392 |
| < = 4 people | 534(38.0) | 176(38.0) | 358(38.0) | |
| 5–8 people | 721(51.4) | 245(52.9) | 476(50.6) | |
| > = 9 people | 149(10.6) | 42(9.1) | 107(11.4) | |
| Desired number of children | | | | 0.000* |
| 0 | 234(16.7) | 95(20.5) | 139(14.8) | |
| 1–2 | 226(16.1) | 86(18.6) | 140(14.9) | |
| 3–5 | 551(39.2) | 164(35.4) | 387(41.1) | |
| >5 | 393(28.0) | 118(25.5) | 275(29.2) | |
| Need for future child | | | | 0.000* |
| No | 234(16.7) | 95(20.5) | 139(14.8) | |
| Yes | 1170(83.3) | 368(79.5) | 802(85.2) | |

Please be aware that the asterisk

(*) in this context denotes the measurements highlighting the difference between Favorable attitudes and Unfavorable attitudes, as determined through crosstabs with Characteristics at a significance level (α) of 0.05. Conversely, absence of the asterisk indicates that the observed difference is not significant.

**Table 2. Differences in attitudes towards contraceptive methods among married couples, Fentale District, Eastern Ethiopia.**

| Distribution (%) | | | |
|---|---|---|---|
| Perception/Attitude | Men (N = 702) | Women (N = 702) | P-value |
| **Perceived susceptibility** | | | |
| I am at risk of unwanted pregnancy if I don't use contraceptive | 180(25.6) | 30(42.9) | 0.000* |
| There is risk of maternal death if unwanted pregnancies happen | 147(20.9) | 211(30.1) | 0.000* |
| **Perceived severity** | | | |
| Unwanted pregnancy can result unsafe abortion | 58(8.3) | 197(28.1) | 0.000* |
| Unsafe abortion can lead to maternal death | 79(11.3) | 259(36.9) | 0.000* |
| Births with close gap can affect the health of a woman and children | 197(27.9) | 175(24.9) | 0.163 |
| Unwanted pregnancy can affect a family economically | 147(20.9) | 160(22.8) | 0.695 |
| **Perceived benefit** | | | |
| Modern contraceptives can prevent pregnancy effectively | 163(23.2) | 249(35.5) | 0.000* |
| Modern contraceptives prevents abortion related consequences | 256(36.5) | 257(36.6) | 0.943 |
| Modern contraceptives help in birth spacing | 64(9.1) | 208(29.6) | 0.000* |
| Some modern contraceptives can prevent HIV/AIDS transmission | 124(17.7) | 88(12.5) | 0.024* |
| **Perceived barrier** | | | |
| Access to modern contraceptives is not too far away | 134(19.1) | 279(39.7) | 0.000* |
| The cost of using modern contraceptives is affordable | 327(46.6) | 243(34.6) | 0.000* |
| To use modern contraceptives is to not fear the side effects | 303(43.2) | 123(17.5) | 0.000* |
| It is convenient for me to use modern contraceptives | 393(56.0) | 267(38.0) | 0.000* |
| **Subjective Norms** | | | |
| Using modern contraceptives is not against my religious tenets | 112(16.0) | 178(25.4) | 0.000* |
| It is not against my culture to use modern contraceptives | 200(28.5) | 242(34.5) | 0.045* |
| My partner approves of using modern contraceptives | 203(28.9) | 264(37.6) | 0.000* |
| **Perceived self-efficacy** | | | |
| I I I feel confident that modern contraceptive will prevent unwanted pregnancy | 228(32.5) | 238(33.9) | 0.851 |
| I I I Would have a confidence to suggest my partner to use modern contraceptive | 150(21.4) | 152(21.7) | 0.720 |
| I a I am confident to ask modern contraceptive methods in health institution | 130(18.5) | 270(38.5) | 0.000* |

Please note that the asterisk

(*) in this context indicates the measurements highlighting the difference between Women's' attitudes and Men's' attitudes, as determined through crosstabs with Perception/Attitude at a significance level (α) of 0.05. Conversely, the absence of the asterisk indicates that the observed difference is not significant.

- Overall, a considerable majority of married couples approve of their partners using modern contraceptives, with a higher percentage reported among women (37.6%) than men (28.9%).

  Perceived Self-efficacy about Family Planning (FP) Services:

- Approximately half of married couples expressed confidence in the ability of modern contraceptives to prevent unwanted pregnancies, with a notably higher percentage reported by women (38.5%) compared to men (18.5%).

- Additionally, only a minority of married couples reported feeling confident in asking for modern contraceptive methods at a health institution, with women (38.5%) outnumbering men (18.5%).

  In Conclusion: The study offers insights into the attitudes and perceptions of married couples regarding family planning services, revealing gender disparities in their views on susceptibility, severity, benefits, barriers, beliefs, and self-efficacy. Notably, variations between men

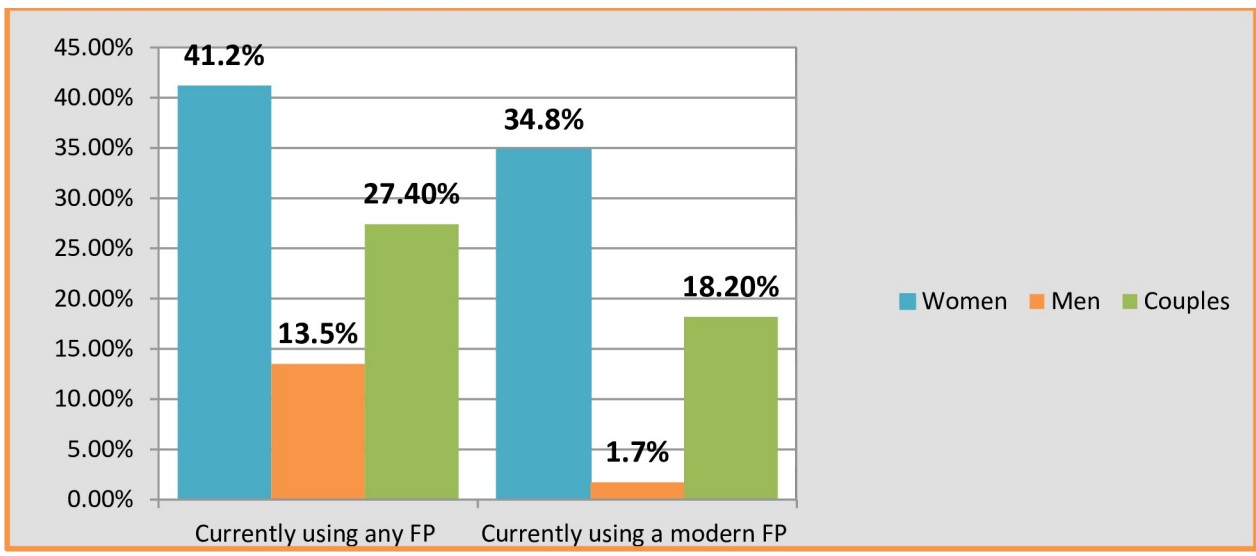

**Fig 2. Current use of modern contraceptives and any types of contraception among couples in Fentale District, Eastern Ethiopia.**

and women are evident in certain aspects, such as the perceived risk of unwanted pregnancy and beliefs about the benefits of modern contraceptives (Table 2).

### 3.3. Contraceptive attitudes and usage among married couples

Fig 2 illustrates the contraceptive usage among married couples based on the survey findings. The figure indicates that 27.4% of couples utilize contraception, revealing a substantial gender disparity: 41.2% of women use contraception compared to 13.5% of men. Furthermore, the figure highlights the use of modern contraceptives, with 18.2% of couples reporting their use. This data also shows a significant gender gap in the use of modern contraceptives, with 34.8% of women using them versus only 1.7% of men (Fig 2).

Fig 3 depicts the attitudes of married couples towards contraception. Among contraception users, 67.8% expressed a positive attitude. However, this varies significantly by gender: 87.9% of women had a positive attitude compared to only 31.9% of men. The analysis also shows that 33% of couples view contraceptive methods positively, but men are more likely to hold unfavorable views (76.4%). For users of modern contraceptives, 52.7% had a positive attitude, with women demonstrating a much higher favorable view (78.5%) compared to men (6.6%). Overall, 33% of married couples have a positive attitude toward contraception, with a higher proportion of women (42.3%) in favor compared to men (23.6%). These significant gender disparities in attitudes and usage underscore the importance of considering gender-specific factors in family planning interventions (P < 0.05) (Fig 3).

### 3.4. Couples' disparities in sources of information on contraceptive methods

Fig 4 illustrates the sources of family planning information received by couples. Approximately 42.5% of couples obtained family planning information, with health extension workers being the primary source. These workers provided information to 42% of husbands and 43% of wives. Mass media, including TV, radio, and newspapers, played a minor role, reaching only 5.6% of individuals. Friends were also a notable source of information, with 28% of husbands and 9% of wives relying on them. Additional sources of information included schools,

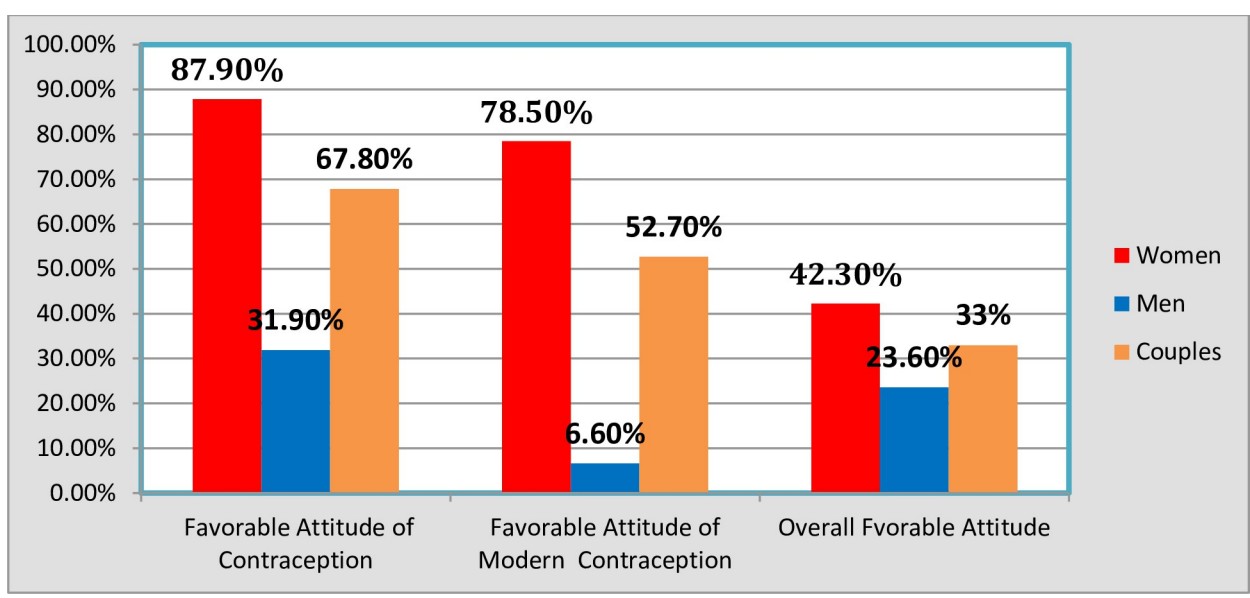

**Fig 3. Favorable attitude of contraception among couples using contraception and comprehensive favorable attitude at study period, Fentale District, Eastern Ethiopia.**

with 15% of husbands and 13% of wives receiving information from this source. Families were also a source, with 15% of husbands and 1% of wives obtaining information from family members. Healthcare providers played a significant role, with 20% of husbands and 22% of wives receiving information from them (Fig 4).

## 3.5. Influence of socio-demographic traits, economic conditions, social factors, and health service access on contraceptive attitudes

Table 3 highlights the factors that influence contraceptive attitudes among married couples. From the 21 variables examined (Table 1), 16 showed significant associations in the bivariate

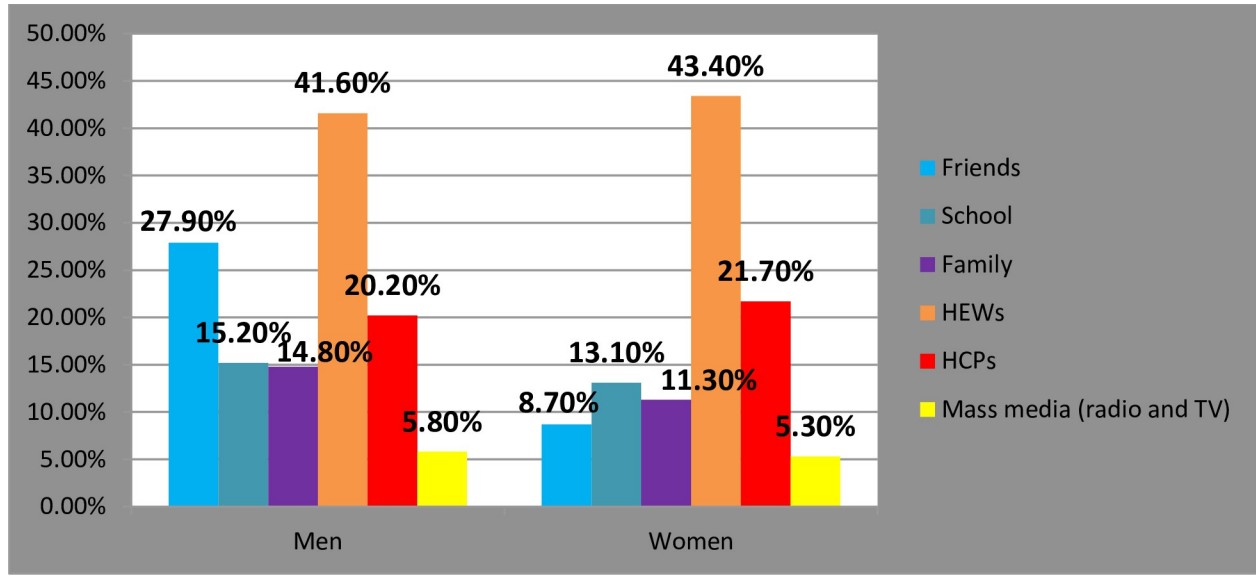

**Fig 4. Distribution of information sources among couples in Fentale District, Eastern Ethiopia, and October 1 to December 25 2021.**

**Table 3. Multivariable logistic regression model for predictors of attitude towards family planning among married couples.** The data covers the period from October 1 to December 25, 2021.

| Characteristics | COR | 95% CI | AOR | 95% CI |
|---|---|---|---|---|
| **Sex** | | | | |
| Women | 1 | | 1 | |
| Men | 0.422 | (0.336–0.53) | 0.494 | (0.321–0.761) |
| **Educational status** | | | | |
| No formal education | 1 | | 1 | |
| Primary | 1.625 | (1.269–2.081) | 1.608 | (1.192–2.169) |
| Secondary & above | 2.014 | (1.450–2.797) | 1.848 | (1.207–2.828) |
| **Religion**\*\* | | | | |
| Muslim | 1 | | 1 | |
| Christian | 2.067 | (1.002–4.266) | 0.892 | (0.357–2.231) |
| **Occupational status** | | | | |
| Nomadic-pastoralist | 1 | | 1 | |
| Business | 1.762 | (1.151–2.697) | 0.852 | (0.496–1.463) |
| Others | 4.698 | (2.049–10.769) | 3.121 | (1.133–8.601) |
| Student | 2.819 | (1.373–5.787) | 2.174 | (0.909–5.199) |
| Agro-Pastoralist | 1.886 | (1.454–2.446) | 1.518 | (1.114–2.068) |
| **Possession of radio** | | | | |
| No | 1 | | 1 | |
| Yes | 2.501 | (1.588–3.941) | 1.824 | (1.040–3.201) |
| **'Possession of mobile phone** | | | | |
| No | 1 | | 1 | |
| Yes | 1.817 | (1.377–2.397 | 1.686 | (1.151–2.469) |
| **Possession of Bank account**\*\* | | | | |
| No | 1 | | 1 | |
| Yes | 0.507 | (0.401–0.642) | 0.778 | (0.527–1.150) |
| **Use the internet**\*\* | | | | |
| No | 1 | | 1 | |
| Yes | 1.921 | (1.227–3.008) | 0.680 | (0.353–1.310) |
| **Couple's Exposure to media** \*\* | | | | |
| Less Frequent | 1 | | 1 | |
| More Frequent | 1.879 | (1.369–2.578) | 1.249 | (0.694–2.246) |
| **Frequency of Migration** | | | | |
| Once | 1 | | 1 | |
| Twice | 0.430 | (0.162–1.145) | 0.430 | (0.139, 1.332) |
| Three | 0.213 | (0.081–0.563) | 0.356 | (0.117–1.089) |
| Four | 0.152 | (0.058–0.400) | 0.251 | (0.082–0.765) |
| Five& more | 0.098 | (0.036–0.264) | 0.218 | (0.069–0.686) |
| Characteristics | COR | (95% CI | AOR | (95% CI) |
| **Family members who migrate mostly**\*\* | | | | |
| Head of the household | 1 | | 1 | |
| Young men | 0.686 | (0.473–0.995) | 1.231 | (0.792–1.913) |
| All family members | 1.296 | (1.022–1.643) | 1.168 | (0.878–1.553) |
| **Place to get Treatment or Where can sick be cured** | | | | |
| At Health Sectors | 1 | | 1 | |
| Traditional healers | 20.807 | (4.716–91.796) | 0.175 | (0.125–0.246) |
| Religious places | 2.608 | (0.595–11.425) | 0.156 | (0.108–0.227) |

*(Continued)*

**Table 3.** (Continued)

| Characteristics | COR | 95% CI | AOR | 95% CI |
|---|---|---|---|---|
| Others | 1.949 | (0.442–8.600) | 0.063 | (0.014–0.292) |
| **Distance from health center\*\*** | | | | |
| < 1 hour | 1 | | 1 | |
| ≥ 1 hour | 0.568 | (0.454–0.711) | 0.931 | (0.705–1.228) |
| **Couple discussion** | | | | |
| Never discussed | 1 | | 1 | |
| Discussed | 3.099 | (2.032–4.728) | 2.403 | (1.143–5.053) |
| **Desired number of additional children\*\*** | | | | |
| 0 | 1 | | 1 | |
| 1–2 | 0.899 | (0.618–1.307) | 1.030 | (0.675–1.569) |
| 3–5 | 0.620 | (0.451–0.853) | 0.967 | (0.696–1.343) |
| >5 | 0.628 | (0.448–0.881) | 0.909 | (0.666–1.241) |
| **Need future child\*\*** | | | | |
| No | 1 | | 1 | |
| Yes | 0.671 | (0.503–0.896) | 0.847 | (0.567–1.265) |

Please note that the double asterisk

(\*\*) in this context indicates a non-significant difference between Favorable attitudes and Unfavorable attitudes, as determined through binary logistic regression with Characteristics at a significance level (α) of 0.05, or Adjusted Odds Ratio (AOR) with a 95% Confidence Interval (CI) including 1. Conversely, the absence of the asterisk or AOR with 95% CI not including 1 indicates that the observed difference is significant.

Note: Predictors demonstrating a significant difference between women and men, with a probability value less than 5%, were included in the multivariate logistic regression. Conversely, those that were not significant at a significance level (α) of 0.05 were excluded from the multivariate logistic regression analysis.

analysis. In the multivariable logistic regression analysis, certain factors, including couples' age, age at first marriage, ethnicity, migration destination, and family size, were not significant. However, eight variables maintained significance: sex of couples, educational status, occupational status, ownership of a radio and mobile phone, migration frequency, treatment location, and couples' discussion of family planning.

Notably, married men were less likely (AOR = 0.494; 95% CI: 0.321–0.761) to have a favorable attitude toward contraceptive methods compared to married women. Conversely, women were 50.6% more likely (AOR = 0.494; 95% CI: 0.321–0.761) to have a favorable attitude compared to men. Couples with a primary education level were 61% (AOR = 1.608; 95% CI: 1.192–2.169) more likely, and those with secondary education and above were 85% (AOR = 1.848; 95% CI: 1.207–2.828) more likely to have a favorable attitude compared to couples with no formal education.

Couples engaged in business were less likely (AOR = 0.852; 95% CI: 0.496–1.463) to have favorable attitudes compared to nomadic pastoralists. Conversely, students, those in other occupations like daily labor and employment, and agro-pastoralists were more likely to have a favorable attitude. Couples with a radio (AOR = 1.824; 95% CI: 1.040–3.201) and mobile phones (AOR = 1.686; 95% CI: 1.151–2.469) were more likely to have favorable attitudes.

Regarding migration frequency, couples who migrated two or more times were less likely to have favorable attitudes compared to those who migrated once. Regarding the place of treatment for sickness, those who sought treatment from traditional healers, and religious places, and those not seeking any treatment were less likely to have favorable attitudes compared to couples treated in health sectors.

Couples who had discussed family planning were two times (AOR = 2.403; 95% CI: 1.143–5.053) more likely to have favorable attitudes.

Regarding the family members who migrate mostly, couples with young men (AOR = 1.231; 95% CI: 0.792–1.913; p = 0.356) and all family members (AOR = 1.168; 95% CI: 0.878–1.553; p = 0.287) were more likely to have a favorable attitude toward modern contraceptive methods than those with the head of the household. Additionally, in assessing media exposure, those with more frequent media exposure (AOR = 1.249; 95% CI: 0.694–2.246; p = 0.459) were more likely to have favorable attitudes toward modern contraceptive methods than those with less frequent exposure. Couples with young men (AOR = 1.231) or all family members (AOR = 1.168) show an increased likelihood of favorable attitudes toward contraception.

However, wide confidence intervals (CIs) spanning 1 and p-values > 0.05 suggest these effects are not statistically significant. Similarly, couples with more frequent media exposure (AOR = 1.249) exhibit an increased likelihood, but wide CI (0.694–2.246) and p-value (0.459) indicate non-significance. In summary, while point estimates suggest effects, wide CIs, and higher p-values imply a lack of statistical significance for these associations. No significant differences were observed for religion, possession of a bank account, use of the internet, couples' exposure to media, family members who migrate mostly, distance from the health center, the desired number of additional children, and the need for future children (See Table 3).

## 4. Discussion

The study addresses the disparities in attitude and utilization of family planning (FP) services among married couples in the pastoralist community of Fentale District, Eastern Ethiopia, highlighting the importance of understanding FP dynamics in this context.

The nomadic lifestyle inherent in pastoralist communities, often necessitated by factors like boundary conflicts and the search for water and grazing lands, poses unique challenges for intervention implementation [45, 46]. Recognizing the nomadic nature of these communities, the study underscores the importance of interventions that are adaptable and tailored to the specific context [47]. Despite challenges related to tracking migration, thorough identification of migration patterns enhances the study's reliability [48].

The novelty of this study lies in its dual focus on both partners within the marital relationship in a pastoralist setting [49]. By examining attitudes and practices toward family planning among couples, it provides a comprehensive view that integrates the perspectives of both men and women, highlighting gender-specific dynamics that influence contraceptive decision-making [50].

When contrasting attitudes towards family planning (FP) and spousal contraceptive practices between the pastoralist Fentale District and non-pastoralist regions of Ethiopia, a significant disparity becomes evident [33].The study highlights considerably lower favorable attitudes towards FP in pastoralist areas, underscoring the pressing need for customized strategies to address the disparities stemming from the nomadic lifestyle [16].

The research resonates with global initiatives such as the Sustainable Development Goals (SDGs), highlighting the importance of promoting positive attitudes towards family planning [51].It echoes the overarching objective of advancing gender equality and facilitating universal access to reproductive health services [17, 52]. By referencing data from indicators from 2017 [53] and situating its findings within the broader national landscape [17], the study underscores the diversity in contraceptive behaviors and attitudes among married couples, underscoring the necessity for interventions tailored to specific regions [17, 54].

The research conducted in the Fentale District of Eastern Ethiopia provides significant insights into family planning dynamics among highly nomadic pastoralist communities.

Despite encountering challenges in engaging this population, the study achieved an impressive 93.8% participation rate among 1496 couples [3, 55].

One major obstacle was determining the timing of their migration to locate grass and water for their livestock. We addressed this challenge by identifying when and where they migrated. This early identification significantly contributed to the high response rate achieved in the study [56]. Distinctive age patterns in first marriage were evident, with men typically marrying at 18 years old and women at 15 years old. These patterns underscore the influence of cultural practices and norms within the community [57].

Additionally, a considerable segment of the population lacked formal education, accounting for 53.8%, while a significant proportion engaged in nomadic-pastoral livelihoods, comprising 64.6% of the community. These findings reflect the distinct lifestyle and occupational characteristics of the community [58].

The Oromo and Muslim community, which comprises the majority, places significant emphasis on cultural foundations in family planning [59, 60].

The nomadic-pastoralist lifestyle presents challenges for disseminating information due to frequent travel [44, 61]. Media challenges affect 87.1% of couples, highlighting socio-economic differences due to gender-based resource disparities [62]. Health preferences vary among the community, with 32.6% of men and 14.5% of women preferring the health sector, while 43.4% opt for traditional healers, and 31.7% choose religious leaders [36, 63].The research conducted in Fentale District, Eastern Ethiopia, reveals essential socio-demographic insights necessary for customizing family planning interventions for nomadic pastoralist communities. It underscores the significance of cultural, educational, and economic factors in shaping effective strategies [3].

Communication barriers impact 93.2% of couples, while the median household size stands at 5 members, with a desire for an additional 3 children, highlighting challenges at the nexus of education and family planning [63–65]. Acknowledging these differences is vital for implementing culturally appropriate family planning interventions in nomadic pastoralist communities [66].

The study investigated attitudes and perceptions towards family planning (FP) services among married couples, revealing notable gender disparities [67]. Women exhibited a higher perceived susceptibility to unwanted pregnancy and maternal mortality, emphasizing their greater concern for FP [68]. Furthermore, women expressed a higher perceived severity of unwanted pregnancy complications, reflecting their heightened awareness of associated risks [69]. While women perceived more benefits of modern contraceptives in preventing pregnancy and its consequences [70], men showed slightly higher positivity towards preventing HIV/AIDS transmission [71]. Access to FP services and affordability were perceived as less significant barriers by women compared to men [72], highlighting disparities in perceived accessibility [73]. Additionally, women were more inclined to believe that using modern contraceptives aligns with religious and cultural principles compared to men [74]. Despite these disparities, a considerable proportion of couples approved of their partners using modern contraceptives, albeit with varying degrees of self-efficacy in contraceptive use and communication [75].

The findings underscore the gender-specific nature of attitudes and perceptions towards FP services among married couples in the pastoralist community [76]. Gender disparities exist across various domains, including perceived susceptibility, severity, benefits, barriers, beliefs, and self-efficacy [77]. These insights emphasize the need for targeted interventions to address disparities and promote equitable access to FP services communities [77], ultimately contributing to improved reproductive health outcomes in pastoralist.

The study reveals notable gender disparities in contraceptive attitudes and usage among married couples in the pastoralist community [3]. While 27.4% of couples reported using

contraception, a substantial difference exists between women (41.2%) and men (13.5%), indicating a higher prevalence among women [54, 78]. Despite overall positive attitudes towards contraception among couples using any form of contraception (67.8%) [79], a significant gender gap is evident, with a higher proportion of women (87.9%) than men (31.9%) exhibiting favorable attitudes [80]. Descriptive analysis highlights variability in attitudes towards contraceptive methods, with only 33% of couples demonstrating a favorable stance [81]. Men, in particular, exhibit a higher prevalence of unfavorable attitudes (76.4%) [82].

Regarding modern contraception, a mere 18.2% of couples reported its usage, with a stark contrast between women (34.8%) and men (1.7%) [70]. Among modern contraception users, a substantial disparity in favorable attitudes is observed, with women (78.5%) far outweighing men (6.6%) [82].

In conclusion, Figs 2 and 3 provide clear insights into the relationship between contraceptive attitudes and adoption, particularly in the context of Fentale District. Fig 2 notably reveals that wives have a higher rate of contraceptive usage than their husbands. Additionally, Fig 3 underscores that wives who use contraception generally hold more positive attitudes toward contraceptive methods than their husbands. Therefore, promoting a favorable attitude towards contraception among couples can effectively enhance contraceptive uptake [43, 82].

Overall, 33% of married couples exhibit a positive attitude towards contraception, with women (42.3%) more inclined than men (23.6%) [16]. The pronounced gender disparity in attitudes and contraceptive usage underscores the importance of addressing gender-specific factors in family planning initiatives [83]. These findings emphasize the need for tailored interventions to bridge the gap and promote equitable access to and acceptance of contraception among married couples in pastoralist communities [84] (P < 0.05).

Couples' disparities in sources of information on contraceptive methods were also highlighted, with health extension workers being the primary source of family planning information [16], followed by friends and healthcare providers. The findings presented in Table 3 shed light on the influence of various socio-demographic factors on attitudes towards family planning (FP) among married couples in the Fentale District.

Notably, gender disparities are evident, with women exhibiting a higher frequency of favorable attitudes toward Family planning compared to men, indicating a need for targeted interventions to address this gap [85]. Education also plays a significant role, with couples having higher education levels showing more favorable attitudes toward Family planning, suggesting the importance of educational initiatives in promoting positive FP attitudes [4].

Religious affiliation appears to influence attitudes, with Muslims displaying a higher percentage of unfavorable attitudes compared to Christians, highlighting the need for culturally sensitive approaches to FP promotion [86]. Furthermore, occupational status, access to communication technology, media exposure, migration patterns, healthcare access, and fertility desires all contribute to variations in FP attitudes [87]. These findings underscore the multifaceted nature of factors shaping FP attitudes and underscore the importance of tailored interventions that address the diverse needs and circumstances of married couples [88] in the Fentale District.

The results presented in Table 3 provide valuable insights into the socio-demographic and reproductive factors influencing attitudes toward contraceptive methods among married couples in the study population. Several significant associations emerged, highlighting the complex interplay between various factors and attitudes towards contraception [78]. Notably, gender emerged as a significant predictor of contraceptive attitudes, with married women being more likely to have favorable attitudes compared to men [83]. This underscores the importance of considering gender dynamics in family planning interventions [80]. Education also played a crucial role, with couples having higher levels of education demonstrating more favorable attitudes towards contraception [16].

Similarly, occupational status influenced attitudes, with certain occupations being associated with greater favorability towards contraceptive methods [16]. Occupationally, couples engaged in business were less likely to have favorable attitudes toward Family planning compared to nomadic pastoralists, while students and those in other occupations were more likely to have favorable attitudes toward Family planning [3].

Access to communication technology, such as radios and mobile phones, emerged as significant predictors of favorable attitudes toward Family planning, suggesting the potential role of media and communication channels in shaping contraceptive beliefs [89]. Additionally, couples who sought treatment from health sectors, as opposed to traditional healers or religious places, were more likely to have favorable attitudes [16, 90]. Furthermore, discussions about family planning within couples were strongly associated with favorable attitudes, highlighting the importance of interpersonal communication in influencing contraceptive decisions [91].

Regarding migration frequency, couples who migrated two or more times were less likely to have favorable attitudes toward Family planning compared to those who migrated once [92]. While certain factors, such as family migration patterns and media exposure, showed trends towards increased likelihood of favorable attitudes toward Family planning [93], the wide confidence intervals and non-significant p-values indicate the need for further research to elucidate their true significance. Overall, these findings underscore the multifaceted nature of factors influencing contraceptive attitudes and emphasize the importance of tailored interventions that address the diverse socio-demographic and reproductive characteristics of married couples.

## 5. Limitations

Limitations to consider include:

1. The study's findings may not be broadly applicable beyond nomadic and agro-pastoralist communities in the Fentale District due to unique socio-cultural dynamics.

2. Challenges in data collection arise from the nomadic lifestyle, making it hard to track individuals consistently and leading to potential information gaps.

3. Language and cultural barriers could hinder effective communication, impacting response accuracy.

4. Reliance on self-reported data may introduce bias, as participants might provide socially desirable responses rather than factual information.

5. Limited female participation and reliance on husbands for information may constrain female autonomy, affecting the accuracy of reproductive health data.

6. Cultural constraints may limit open discussions on certain topics, potentially leading to underreporting or hesitancy in sharing family planning information.

7. Findings may reflect a specific point in time due to the nomadic nature of the community, lacking long-term trend insights.

8. The dynamic pastoralist lifestyle suggests evolving attitudes and practices not entirely captured in the study.

## 6. Conclusion

In summary, the study underscores the necessity of customized interventions for pastoralist communities in Fentale District, Eastern Ethiopia, with a particular emphasis on couple-centered health education, especially targeting men. It is imperative to develop comprehensive

strategies that engage both genders to address the disparities in family planning practices. The study identifies critical areas for intervention, including gender dynamics, educational attainment, and occupational status, ownership of electronic devices, migration patterns, treatment preferences, and discussions about family planning. Collaborating with religious leaders and community influencers is vital to enhance the impact of interventions within the cultural framework. The establishment of mobile clinics and educational programs, delivered by culturally adept educators, is essential for creating interventions that are suited to the distinctive lifestyle of pastoralist communities. By tackling the gaps in attitudes, adoption, and disparities between couples, these interventions can cater to the specific requirements of the pastoralist community and support broader health goals.

The study also brings to light the gender differences in perceptions of family planning, with women recognizing greater risks and benefits than men. Variables such as education, occupation, resource accessibility, media influence, migration tendencies, and healthcare utilization significantly shape attitudes toward contraceptive methods among married couples. A deep understanding of the intricate relationship between socio-demographic factors and reproductive experiences is crucial for crafting effective and culturally responsive family planning interventions that are tailored to the unique needs of pastoralist communities. These interventions aim to foster positive attitudes and increase the adoption of contraceptive methods, ultimately promoting reproductive health and well-being in these communities.

## 7. Policy and practice recommendations

Based on our findings, the following policy recommendations are proposed to enhance family planning (FP) interventions in pastoralist communities:

**Culturally Tailored Interventions:** Develop FP programs that respect the cultural norms and nomadic lifestyle of pastoralist communities, ensuring interventions are acceptable and resonate with their values.

**Education and Awareness Programs:** Implement targeted education and awareness initiatives on FP and reproductive health, tailored to the literacy levels and learning styles of pastoralists, to empower them with decision-making skills.

**Mobile and Radio Communication:** Utilize mobile phones and radios to disseminate FP information and services, supporting media campaigns that address misinformation and promote FP awareness.

**Healthcare Access and Training:** Strengthen healthcare infrastructure in pastoralist areas and train providers to deliver comprehensive FP services, ensuring accessibility for nomadic populations.

**Gender-Sensitive Approaches:** Address gender disparities in FP attitudes and usage by incorporating interventions that tackle decision-making disparities and empower women.

**Monitor Migration Patterns:** Implement tracking systems for migration flows to enhance program execution and community involvement.

**Community Engagement and Participation:** Involve community leaders, local health workers, and pastoralist families in the design and implementation of FP programs to ensure they meet the community's needs.

**Integration with Global Initiatives:** Align FP services with global initiatives like the SDGs to promote gender equality and universal access to reproductive health, integrating FP with other health and development programs.

These recommendations aim to address the unique challenges of pastoralist communities, fostering effective strategies to improve reproductive health outcomes and equitable access to FP services.

## Acknowledgments

The authors extend their appreciation to our supervisors for their valuable guidance, the diligent data collectors, and the respondents for their participation. Gratitude is also expressed to the Oromia Regional Health Bureau, Zonal authorities, and Fentale District administration for their facilitation and support throughout the study. Additionally, the authors acknowledge the 'Fantale Woreda Socio-economic Office, 2020' for providing information on the number of households in Fentale.

## Author Contributions

**Conceptualization:** Sileshi Garoma, Tefera Belachew.

**Data curation:** Sena Adugna Beyene, Tefera Belachew.

**Formal analysis:** Sena Adugna Beyene.

**Funding acquisition:** Sena Adugna Beyene, Sileshi Garoma, Tefera Belachew.

**Investigation:** Sena Adugna Beyene, Sileshi Garoma, Tefera Belachew.

**Methodology:** Sena Adugna Beyene, Sileshi Garoma, Tefera Belachew.

**Project administration:** Sena Adugna Beyene, Sileshi Garoma, Tefera Belachew.

**Resources:** Sena Adugna Beyene, Sileshi Garoma, Tefera Belachew.

**Software:** Sena Adugna Beyene.

**Supervision:** Sena Adugna Beyene.

**Validation:** Sena Adugna Beyene, Sileshi Garoma, Tefera Belachew.

**Visualization:** Sena Adugna Beyene, Sileshi Garoma, Tefera Belachew.

**Writing – original draft:** Sena Adugna Beyene.

**Writing – review & editing:** Sena Adugna Beyene, Sileshi Garoma, Tefera Belachew.

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
