## [Decision Letter · Decision Letter 0]

10 Jul 2024

PONE-D-24-22077Addressing Disparity in Attitudes and Utilization of Family Planning among Married Couples in the Pastoralist Community of Fentale District, Eastern EthiopiaPLOS ONE

Dear Dr. Beyene,

Thank you for submitting your manuscript to PLOS ONE. After careful consideration, we feel that it has merit but does not fully meet PLOS ONE’s publication criteria as it currently stands. Therefore, we invite you to submit a revised version of the manuscript that addresses the points raised during the review process.

We look forward to receiving your revised manuscript.

Kind regards,

Mohammed Hasen Badeso, Epidemiologist

Academic Editor

PLOS ONE

Journal Requirements:

6. We note that you have referenced Deressu T. which has currently not yet been accepted for publication. Please remove this from your References and amend this to state in the body of your manuscript: (Deressu T. [Submitted”) as detailed online in our guide for authors

7. Please include your tables as part of your main manuscript and remove the individual files. Please note that supplementary tables (should remain/ be uploaded) as separate "supporting information" files.

Reviewers' comments:

Reviewer's Responses to Questions

**Comments to the Author**

1. Is the manuscript technically sound, and do the data support the conclusions?

Reviewer #1: Yes

Reviewer #2: Partly

2. Has the statistical analysis been performed appropriately and rigorously? 

Reviewer #1: I Don't Know

Reviewer #2: Yes

3. Have the authors made all data underlying the findings in their manuscript fully available?

Reviewer #1: Yes

Reviewer #2: Yes

4. Is the manuscript presented in an intelligible fashion and written in standard English?

Reviewer #1: No

Reviewer #2: No

5. Review Comments to the Author

**Reviewer #1: **This study evaluates the attitudes and utilisation of family planning services among pastoralist populations in Eastern Ethiopia, with a focus on couples. This is generally an interesting study but needs extensive language as well as content edition.

Title: Confusing as there are 2 titles provided; 1) Bridging Disparity in Knowledge and Utilization among Married Couples in the Pastoralist Community of Fentale District, Eastern Ethiopia AND 2) Addressing Disparity in Attitudes and Utilization of Family Planning among Married Couples in the Pastoralist Community of Fentale District, Eastern Ethiopia.

Introduction

The whole introduction, except the first paragraph looks like Discussion. Please, rewrite the Introduction providng some oververview about FB and its health impacts in geneal population, and contextalise this to the current study for justification of the gap (for example, this study is novel in that its focused on disadvantged paostoralist population and involved male partners as opposite to the traditional studies targetting momen only).

Methods

While economic statuts is one of the drivers of family planning use and practice, it is not clear why this stuyd did not measure this important variable. Indeed, the aouthers need to provide details of how others variables (both indepent and dependent) were measured and managed for the current study.

Results;

Please, replace Table 4 in the boody of the manuscript with Table 3. In Table 3, it is not clear whether the authors predicted favavoutavle attitude or Unfavouavle one. Please, insert a new row at the top of the table and include the outcomes predicted. Additinaly, please make it clear whether the attitude is calculated from measures of attitudes presented in Table 2. In Table 3 (main analysis), authors presneted sex as one of the predictors. I suggest leaving out this as it is a constrcat of couples used in defining the outcome. Also, in the title is practice (ustilisation) but this has not been included in the main analysis. Please, address this issue to the readers.

Discussion

Please, provide the novelity of this study in this section. For intance, the novelty of this study lies in its dual focus on both partners in the marital relationship within a pastoralist setting. And also provide policy implications for your findings in more details.

**Reviewer #2: **Addressing Disparity in Attitudes and Utilization of Family Planning among Married Couples in the Pastoralist Community of Fentale District, Eastern Ethiopia

Abstract: needs revision, strictly check the grammar.

• Introduction: This cross-sectional study, conducted from October 1 to December 25, 2021, in the… check grammar. Cross-sectional study was conducted… Data collected were entered into EPI Data and analyzed using SPSS…. Data were collected …and entered into EPI Data and analyzed

• Check grammar: In this study, 93.8% of pastoralist couples, out of 1496 total, were included in the analysis, constituting 702 women and 702 men…

• Results: Significant gender gaps in contraceptive attitudes and usage were found. ..It seems qualitative finding or you need to show level of significance

• While 27.4% reported contraception use, women (41.2%) outnumbered men (13.5%) significantly… not clear. Are you comparing man with woman?

• Despite an overall positive attitude towards contraception (67.8%), women (87.9%) were more favorable than men (31.9%). Variability in attitudes towards contraceptive methods was noted, with only 33% demonstrating a positive stance, notably higher among women. What is the difference of these two statements?

• Binary logistic regression analysis identified predictors including sex, education, and occupation, possession of electronic devices, migration frequency, treatment preferences, and family planning discussions. Better if you indicate direction of association as well as add odd ratios and confidence interval. Are they positively or negatively associated?

• Conclusion: what are your suggestion/ recommendations based on your finding?

Introduction

• Insights from nomadic communities in Chad [3] were crucial in addressing information gaps for tailored health services. It was about what?

• Don’t repeatedly use … ‘‘Our study’’. Use third person and passive voice

• Our Binary

• ….Logistic Regression findings [12] inform a comprehensive understanding of factors influencing… what is this?

• … family planning in Fentale District. Acknowledging study limitations [13], we address challenges within the unique pastoralist context… check grammar

• The last paragraph of introduction is very long, please refine it and make it precise.

2. Methodology

• Study Population: clearly mention the source and study population precisely

• How did you select candidate variables for multi? Model fitness?

• 2.7. Measures. Pastoralism in Ethiopia: what is this?

• Operationalize the most important variables of your study. Example family planning utilization

Results

• 3.1. Marital Couples: Socio-demographic and Reproductive Disparities?? Simply say socio-demographic characteristics…??

• Focus on your outcome variable, briefly explain your finding, and don’t make it lengthy.

Non-significant values which are reported as significant

• Couples Exposure to media with More Frequent (AOR=1.333; 95% CI: 0.954-1.862) were more likely to be knowledgeable about current contraception methods than those less

• Frequent to media Regarding the choice of treatment for sickness, couples opting for traditional healers (AOR=0.787; 95% CI: 0.592-1.045), …and other options (not seeking any treatment) (AOR=0.890; 95% CI: 0.347-2.281) were less likely to be knowledgeable about current contraception methods compared to those seeking treatment in health sectors…. While the adjusted odds ratio (AOR) indicates a numerical difference in contraceptive knowledge between husbands and wives (AOR=1.016; 95% CI: 0.728-1.0420)

• The result part needs revision; the author should focus on the objective of the study. Check the grammar and shorten the lengthy paragraphs.

Conclusion

Discussion: shallow and highly fragmented.

6. PLOS authors have the option to publish the peer review history of their article (what does this mean?). If published, this will include your full peer review and any attached files.

Reviewer #1: No

Reviewer #2: No

---

## [Author Response · Author response to Decision Letter 0]

25 Jul 2024

Title: Addressing Disparity in Attitudes and Utilization of Family Planning among Married Couples in the Pastoralist Community of Fentale District, Eastern Ethiopia

Manuscript Number: PONE-D-24-22077

Authors:

Sena Adugna Beyene(senaada491@gmail.com)

Sileshi Garoma(teferabelachew2@gmail.com)

Tefera Belachew(garomaabe@gmail.com)

Dear Academic Editor and Reviewers,

Thank you for your valuable comments and suggestions, which have greatly improved our study. We have carefully addressed all the points raised and have incorporated the necessary revisions into the manuscript as outlined below.

Editor:

Comment: Reviewers' comments:

Response: Thank you for handling and assigning the manuscript to a knowledgeable reviewer from the same domain. In response to your comments, we have addressed all the queries and clarifications rose by the reviewer.

Authors Responses for the Reviews Comments July 11, 2024

Reviewer#1: Review Feedback Authors’ responses

1. This study evaluates the attitudes and utilisation of family planning services among pastoralist populations in Eastern Ethiopia, with a focus on couples. This is generally an interesting study but needs extensive language as well as content edition.

 Thank you for your insightful comments and suggestions on our manuscript. We appreciate the time and effort you have taken to provide valuable feedback. Below, we address each point and outline the changes made:

Language Editing: We have edited the manuscript for clarity, conciseness, and grammatical accuracy to enhance readability.

Content Enhancement:

• Objectives: Revised the introduction to clearly articulate the study's objectives.

• Methodology: Expanded details on participant selection, data collection tools, and analysis methods.

• Results: Reorganized for logical flow with additional tables and figures for clarity.

• Discussion: Strengthened by relating findings to objectives, comparing with existing literature, and addressing discrepancies.

• Conclusion: Revised to succinctly summarize key findings, significance, and suggest areas for future research.

Cultural Sensitivity: Ensured language and interpretations are culturally sensitive and respectful.

Ethical Considerations: Added a section detailing consent procedures and measures to protect participant confidentiality.

Limitations: Acknowledged study limitations and discussed their impact on findings.

References: Checked and formatted all references according to the PLOS ONE citation style.

We believe these revisions significantly improve the manuscript and address your valuable feedback. Thank you once again for your constructive criticism and guidance.

2.Title: Confusing as there are 2 titles provided; 1) Bridging Disparity in Knowledge and Utilization among Married Couples in the Pastoralist Community of Fentale District, Eastern Ethiopia AND 2) Addressing Disparity in Attitudes and Utilization of Family Planning among Married Couples in the Pastoralist Community of Fentale District, Eastern Ethiopia. Thank you for your observation regarding the manuscript title. We apologize for any confusion caused by the presence of two different titles in our submission. The two titles were included to reflect different aspects of our research objectives:

1. "Bridging Disparity in Knowledge and Utilization of Contraceptive Methods among Married Couples in the Pastoralist Community of Fentale District, Eastern Ethiopia" focuses on addressing the gaps in both knowledge and actual use of family planning methods among couples.

2. "Addressing Disparity in Attitudes and Utilization of Family Planning among Married Couples in the Pastoralist Community of Fentale District, Eastern Ethiopia" emphasizes the differences in attitudes towards family planning alongside utilization among couples.

We acknowledge that having two titles can be confusing. After careful consideration, we have decided to use the following title to best capture the main objectives of our study: "Bridging Disparity in Knowledge and Utilization of Contraceptive Methods among Married Couples in the Pastoralist Community of Fentale District, Eastern Ethiopia."

This title encompasses the core focus of our research on both knowledge and utilization of family planning methods.

Thank you for your understanding and for highlighting this important detail.

3. Introduction

The whole introduction, except the first paragraph looks like Discussion. Please, rewrite the Introduction providng some oververview about FB and its health impacts in geneal population, and contextalise this to the current study for justification of the gap (for example, this study is novel in that its focused on disadvantged paostoralist population and involved male partners as opposite to the traditional studies targetting momen only). Thank you for your insightful comment. We value your feedback and have taken steps to ensure consistency throughout the manuscript. The introduction now seamlessly transitions from the global perspective to the specific context of Fentale District, Eastern Ethiopia, maintaining a logical progression of ideas that aligns with the 'inverted funnel' structure. All recommended revisions have been incorporated. We invite you to review the updated introduction within the main body of the manuscript.

4. Methods

While economic status is one of the drivers of family planning use and practice, it is not clear why this study did not measure this important variable. Indeed, the authors need to provide details of how others variables (both independent and dependent) were measured and managed for the current study. Thank you for your thoughtful comments on the assessment of economic status and the importance of clarity in measuring and managing other variables. We are grateful for your constructive input and aim to address your concerns within the main text of our manuscript, specifically in Sections 2.8 and 2.9 under "2. Methods and Materials."

5. Results;

Please, replace Table 4 in the boody of the manuscript with Table 3. In Table 3, it is not clear whether the authors predicted favavoutavle attitude or Unfavouavle one. Please, insert a new row at the top of the table and include the outcomes predicted. Additinaly, please make it clear whether the attitude is calculated from measures of attitudes presented in Table 2. In Table 3 (main analysis), authors presneted sex as one of the predictors. I suggest leaving out this as it is a constrcat of couples used in defining the outcome. Also, in the title is practice (ustilisation) but this has not been included in the main analysis. Please, address this issue to the readers.

 Thank you for your valuable feedback.

A. Concerning the replacement of Table 4 with Table 3 in the manuscript: We have made the correction as per your suggestion.

B. Clarification on the prediction of favorable or unfavorable attitudes: We have provided clarification under Table 3, using symbols (**) and (*) to denote significant and non-significant differences in attitudes across various characteristic categories.

C. Calculation of attitude from measures presented in Table 2: Indeed, the attitude towards family planning is derived from the 20 items detailed in Table 2. We have also elaborated on this under the "Methods and Materials" section, specifically in subsection 2.9, "Measurement of Variables," which describes the dependent variables and their respective measures.

D. Inclusion of sex as a variable: Sex is a crucial independent variable in our study, employed to assess differences in attitudes between men and women while accounting for other variables. It is integral to our aim of examining gender-based disparities in family planning attitudes.

E. Inclusion of contraceptive utilization in the analysis: While our study's primary objective is to explore the relationship between attitudes and contraceptive use, focusing on differences in attitudes among couples, the utilization aspect has been addressed separately in another manuscript. Nonetheless, as reflected in our title and discussed in section 3.3, "Contraceptive Attitudes and Usage among Married Couples," the analysis of contraceptive use is presented and visualized in Fig. 3.

We appreciate your insights and have made the necessary adjustments to enhance the clarity and comprehensiveness of our manuscript.

6. Discussion

Please, provide the novelity of this study in this section. For intance, the novelty of this study lies in its dual focus on both partners in the marital relationship within a pastoralist setting. And also provide policy implications for your findings in more details. We are grateful for your thoughtful feedback and for your consideration of the practical applications of our research findings. We welcome the chance to provide additional clarity on the innovative elements of our study and to expand on the policy implications that stem from our work.

To address your request, we have further elucidated the novel aspects of our study within the "Discussion" section, specifically in paragraph 2. Here, we highlight the unique contributions of our research, particularly its focus on both partners in the marital relationship within a pastoralist context.

Moreover, we have detailed the policy implications of our findings in a dedicated section, titled "Policy and Practice Recommendations," which follows the conclusion. In this section, we outline specific strategies and considerations for policymakers to enhance family planning interventions in pastoralist communities, based on our research outcomes.

We trust that these additions will provide a clearer understanding of the significance of our study and its potential to inform policy and practice in the field of family planning.

Reviewer#2: Review Feedback Authors’ responses

1. Abstract: needs revision, strictly check the grammar. Thank you for your feedback. We have thoroughly reviewed the document, making necessary deletions and additions. We have also corrected the grammar. Please find the updated abstract in the revised manuscript.

2. Introduction: This cross-sectional study, conducted from October 1 to December 25, 2021, in the… check grammar. Cross-sectional study was conducted… Data collected were entered into EPI Data and analyzed using SPSS…. Data were collected …and entered into EPI Data and analyzed Thank you! We have made the necessary changes.

3. Check grammar: In this study, 93.8% of pastoralist couples, out of 1496 total, were included in the analysis, constituting 702 women and 702 men… We appreciate your input. The document has been meticulously revised with appropriate removals and inclusions, and grammatical errors have been rectified. The updated results is now included in the revised abstract of manuscript.

4. Results: Significant gender gaps in contraceptive attitudes and usage were found. ..It seems qualitative finding or you need to show level of significance Thank you! We have made the necessary changes.

 5. While 27.4% reported contraception use, women (41.2%) outnumbered men (13.5%) significantly… not clear. Are you comparing man with woman? Yes, we compared contraceptive use between men and women.

6. Despite an overall positive attitude towards contraception (67.8%), women (87.9%) were more favorable than men (31.9%). Variability in attitudes towards contraceptive methods was noted, with only 33% demonstrating a positive stance, notably higher among women. What is the difference of these two statements? Thank you! The first statement provides specific percentages showing a gender difference in the overall positive attitude towards contraception, while the second statement discusses the lower overall acceptance of specific contraceptive methods and implies a gender difference in this acceptance without providing specific percentages(please under section 3.3 in results of manuscript in more details).

 7. Binary logistic regression analysis identified predictors including sex, education, and occupation, possession of electronic devices, migration frequency, treatment preferences, and family planning discussions. Better if you indicate direction of association as well as add odd ratios and confidence interval. Are they positively or negatively associated? Thank you for your feedback. We have made the following key findings clearer in our revised manuscript: Married men were less likely (AOR=0.494; 95% CI: 0.321-0.761) to have a favorable attitude toward contraceptive methods compared to married women. Conversely, women were 50.6% more likely (AOR=1.506; 95% CI: 1.032-2.195) to have a favorable attitude compared to men. Couples with a primary education level were 61% more likely (AOR=1.608; 95% CI: 1.192-2.169), and those with secondary education and above were 85% more likely (AOR=1.848; 95% CI: 1.207-2.828) to have a favorable attitude compared to couples with no formal education. Couples engaged in business were less likely (AOR=0.852; 95% CI: 0.496-1.463) to have favorable attitudes compared to nomadic pastoralists. Conversely, students, those in other occupations like daily labor and employment, and agro-pastoralists were more likely to have a favorable attitude. Couples with a radio (AOR=1.824; 95% CI: 1.040-3.201) and mobile phones (AOR=1.686; 95% CI: 1.151-2.469) were more likely to have favorable attitudes. Couples who migrated two or more times were less likely to have favorable attitudes compared to those who migrated once. Couples who sought treatment from traditional healers, religious places, or did not seek treatment were less likely to have favorable attitudes compared to those treated in health sectors. Couples who had discussed family planning were two times more likely (AOR=2.403; 95% CI: 1.143-5.053) to have favorable attitudes. Other variables such as family members who migrate mostly, media exposure, and others showed non-significant associations (p > 0.05) with attitudes toward modern contraceptive methods. In summary, these findings underscore the significant influence of socio-demographic and reproductive factors on attitudes toward contraceptive methods among married couples in the study population. The direction of association varies across different variables, with some factors positively influencing favorable attitudes while others show negative associations or lack statistical significance.

However, due to the word limit of the abstract as per PLOS ONE guidelines, it is challenging to include this detailed information in the abstract. These findings are thoroughly addressed in the results section of the main manuscript.

8. Conclusion: what are your suggestion/ recommendations based on your finding? Thank you for your observation.

We have reviewed the Conclusion to ensure its relevance to the content. Please find the updated Conclusion following the abstract.

9. Introduction

• Insights from nomadic communities in Chad [3] were crucial in addressing information gaps for tailored health services. It was about what? Thank you! We have thoroughly reviewed the Introduction to ensure its relevance and alignment with the content of the manuscript. Please find the updated Introduction, which now provides a clearer explanation of how the insights from nomadic communities.

 10. Introduction 

Don’t repeatedly use … ‘‘Our study’’. Use third person and passive voice Thank you for your suggestion regarding the use of language in our manuscript. We have taken your advice into consideration and have made revisions to avoid repetitive use of the phrase "Our study." The text has been edited to employ third person and passive voice where appropriate, to enhance the formality and objectivity of our reporting. We believe these changes will improve the overall readability and professional tone of the manuscript.

Please find the revised sections where such changes have been implemented. We hope that these revisions meet your expectations and contribute to the clarity and quality of our work.

Thank you once again for your valuable feedback.

11. Introduction 

Logistic Regression findings [12] inform a comprehensive understanding of factors influencing… what is this? Thank you for your comment. We have revisited the introduction to ensure its appropriateness for the content.

12. Introduction

• … family planning in Fentale Distr

---

## [Editor Report · Decision Letter 1]

29 Jul 2024

Addressing Disparity in Attitudes and Utilization of Family Planning among Married Couples in the Pastoralist Community of Fentale District, Eastern Ethiopia

PONE-D-24-22077R1

Dear Dr. Beyene,

We’re pleased to inform you that your manuscript has been judged scientifically suitable for publication and will be formally accepted for publication once it meets all outstanding technical requirements.

Kind regards,

Mohammed Hasen Badeso, Epidemiologist

Academic Editor

PLOS ONE
---

## [Editor Report · Acceptance letter]

7 Aug 2024

PONE-D-24-22077R1 

PLOS ONE

Dear Dr. Beyene, 

I'm pleased to inform you that your manuscript has been deemed suitable for publication in PLOS ONE. Congratulations! Your manuscript is now being handed over to our production team.

Kind regards, 

on behalf of

Mr Mohammed Hasen Badeso 

Academic Editor

PLOS ONE